# Environmental Factors Affecting the Reproductive Efficiency of Italian Simmental Young Bulls

**DOI:** 10.3390/ani12182476

**Published:** 2022-09-19

**Authors:** Francesca Corte Pause, Martina Crociati, Susy Urli, Maurizio Monaci, Lorenzo Degano, Giuseppe Stradaioli

**Affiliations:** 1Department of Agricultural, Food, Environmental and Animal Sciences, University of Udine, Via Delle Scienze 206, 33100 Udine, Italy; 2Department of Veterinary Medicine, University of Perugia, Via S. Costanzo 4, 06126 Perugia, Italy; 3Centre for Perinatal and Reproductive Medicine, University of Perugia, 06129 Perugia, Italy; 4Italian Association of Simmental Breeders (ANAPRI), Via Ippolito Nievo 19, 33100 Udine, Italy

**Keywords:** bovine, bull, semen, scrotal circumference, temperature humidity index, breeding soundness evaluation

## Abstract

**Simple Summary:**

Bull fertility is an important contributor of the herd reproductive success, especially in natural service mating systems. Even if the BBSE (Bulls Breeding Soundness Evaluation) is a rapid and cost-effective screening, it is unable to evaluate all the parameters involved in fertility decline, especially those related to genetics and environment. Based on this, the objectives of the study were (i) to define the semen characteristics of young Italian Simmental bulls, (ii) to evaluate the relationship between scrotal circumference (SC), age, and semen characteristics and (iii) to relate the climatic conditions during the entire spermatogenesis to variations in spermatozoa motility and morphology. Concerning the evaluation of SC, this is the first report on Italian Simmental breed; its increase was linked with the concentration and number of total spermatozoa. Considering the effect of the weather changes, the study confirmed the impact of the increasing temperature and humidity on spermatozoa motility and morphology, accordingly with the moment in which the heat stress occurred. Therefore, considering the importance of fertility, wider knowledge of the factors influencing these parameters will help maintain/improve the reproductive capability of the animals and finally, for the producer, the profitability of his herd.

**Abstract:**

The objectives of the study aimed to evaluate the effect of weather conditions and scrotal circumference (SC) on standard semen characteristics of Italian Simmental young bulls (*n* = 577), all raised in the same performance station and sampled by the artificial vagina (AV) method. Considering the increasing SC, the results showed a significant increase in quantitative semen parameters (*p* < 0.05 and *p* < 0.0001, for sperm concentration and total number of spermatozoa, respectively); for every extra centimeter of SC, 17.5 × 10^6^ spermatozoa/mL and 0.102 × 10^9^ of total spermatozoa were produced. The age of the animal at semen collection (395 and 465 days) had similar positive effects. The effect of the average temperature humidity index (THI limits ≤ 40 and ≥70) in the previous 60, 30, and 10 days before the semen collection was also considered. Sixty days before the semen collection, the increasing THI increased both primary (*p* < 0.0001) and secondary (*p* < 0.0001) abnormalities while the percentage of morphologically normal spermatozoa decreased (*p* < 0.0001). Thirty and ten days before collection, the same effect on morphological traits was maintained, but total and progressive motility was also influenced (*p* < 0.01) with an unexpected increasing pattern. Thus, environmental conditions can influence semen quality during the entire spermatogenesis and results can guide future research on this breed.

## 1. Introduction

Generally, fertility is described as the ability to conceive and produce viable offspring [1].

Reproductive failures such as low conception and pregnancy rates in artificially inseminated herds are dependent both on female and on bull subfertility, with consistent detriment of the farm net return [2]. Moreover, in natural service mating systems, male fertility is one of the most important contributors to the herd reproductive success [3].

The Bulls Breeding Soundness Evaluation (BBSE) includes physical and reproductive tract examinations (including the measurement of the scrotal circumference), the collection and analysis of a semen sample, and in some cases, a libido/mating ability test to estimate the potential fertility of males; according to a recently published Spanish guide, ultrasound evaluation and sanitary procedures could be added [4,5,6]. In fact, both the quantity and quality of semen are affected by a complex interaction of factors such as genetics, environmental conditions, and management, together with the physiological status of the bull [7,8]. This is particularly true for young bulls that frequently suffer from immaturity, and which are increasingly being used for the early production of ejaculates destined for the artificial insemination (AI) industry, especially in view of the modern genomic selection scheme; in this view, enhancing early-life nutrition has been recognized as a key-element in fastening puberty and improving semen quality [9].

Great importance is given to the selection of the bulls with the largest scrotal circumferences [10]. The SC is the most accurate predictor of testicular size, weight [11], semen quality, overall potential bull fertility [9,12,13,14,15], with an anticipated age at puberty and pregnancy rate of female progeny [16,17,18]. Differently from the other reproductive parameters, SC is medium-high heritable [4,19]; thus, the selection for these desirable traits will improve animals’ reproductive performance [20,21,22].

Moreover, due to the widespread use of frozen-thawed semen, considerable attention is also given to semen characteristics, such as total and progressive motility and presence and type of sperm abnormalities: these are known to influence the semen suitability to freezing and then its fertilizing capacity [23,24]. Despite genetics and handling (for example bull handler and semen collector), testes and all semen characteristics are also influenced by climatic seasonal variations at any time of spermatogenesis (estimated to last around 60 days in bull). Seasonal effects involve several factors such the following: temperature, humidity, photoperiod, and food quality [7], and, for this reason, the exact mechanisms by which seasonal variation influences semen quality have not been unanimously determined. For instance, Snoj et al. [25], who analyzed semen characteristics of four different breeds (Holstein, Brown-Swiss, Charolaise, and Limousine bulls) with an average age varying from 12 months to more than 84 months, reported a higher volume and a greater number of total spermatozoa during summer, but their study was conducted in Slovenia, in a temperate climate (average temperatures ranging from 1.1 °C in winter to 20.3 °C in summer). On the contrary, Bhakat et al. [26] described a lower percentage of motile spermatozoa, a lower sperm concentration and a larger percentage of abnormal sperm in a study performed in Karan Friesian bulls, with an average age varying from 24 to 37 months, in India (average temperatures ranging from 12.7 °C in winter to 31.8 °C in summer). However, the influence of the genotype in this latter breed, which has originated from the crossing of Holstein (*Bos taurus*) and Tharparkar breeds (*Bos indicus*) in a tropical climate adaptation is difficult to compare. The report published by Sylla et al. [27] was carried out in Italy in beef breeds (Chianina, Marchigiana and Romagnola) in the Mediterranean climate; the authors observed that summer months were associated with depression of semen characteristics. Studies that considered environmental conditions often evaluated only the effect on the day of semen collection, ignoring the long-term consequences of extreme weather on the entire spermatogenic cycle [28,29]; others, which also considered long-term interactions, only used temperature. The THI, which combines temperature and humidity, has been introduced as a better indicator of thermal stress in cattle because of its objectively described threshold levels [30].

This study was conducted on bulls from the Italian Simmental breed, which were evaluated in the same genetic center, from 2014 to 2021.

The objectives of this study were: (i) to establish mean values for seminal characteristics of Italian Simmental young bulls raised in the same performance station, (ii) to evaluate a linkage between testicular dimensions and the parameters considered during semen analysis, and (iii) to relate the effect of the average temperature humidity index-THI in the previous 60, 30, and 10 days before the semen collection on spermatozoa concentration, motility, and morphology.

## 2. Materials and Methods

### 2.1. Animals

The procedures used in this investigation were approved by the Committee for Animal Welfare, Udine University (Prot. OPBA N. 01/2014).

The animals involved in this research derived from the best young calves, selected each year based on morphological traits and the sires’ genetic index by the technicians of the Italian Association of Simmental Breeders.

In order to measure the growth performance and meat production attitude, the young calves were first introduced into the testing station located in Fiume Veneto (Friuli Venezia Giulia, in the North-East of Italy, 45°91′90″10 N, 12°72′31″70 E). Calves, tested negative for brucellosis, tuberculosis, bovine leukemia virus, infectious bovine rhinotracheitis, and bovine virus diarrhea, were 25 ± 10 days old when they arrived at the testing station and fed by bucket until 60 days of age with free access to water, hay, and weaning pellets. At the age of about 3 months, they were divided into homogeneous groups (5–6 animals) and gradually prepared for the period of performance-test, which normally starts when they are 5 months old and ends when they are 12 months old. They were fed with a total mixed ration (TMR) distributed ad libitum twice a day, which contained: ground corn, corn silage, sunflower and rapeseed meals, wheat straw, barley, dried beet pulp, wheat bran, soybean meal, and a mineral and vitamin mix. In addition, the diet chemical composition was defined as follows: 64.9% dry matter (DM), 13.5% crude protein, 6.3% ash, and 35.9% fiber (Neutral detergent fiber, NDF) [31]. During this period, they were also weighed and measured monthly. Then, morphologically satisfactory bulls were moved to individual boxes. Only at this point did they undergo semen collection and analysis.

At the end of the performance period, all bulls were subjected to measurement of some somatic traits (stature, abdominal depth, sacrum height, chest circle, chest deep, rump length, rump width) and for scrotal circumference. Furthermore, they were also morphologically evaluated for frame, muscling, conformation of feet and legs, and Body Condition Score (BCS). At the time of morphological evaluation, the average age of the bulls was 387.2 ± 11 days (age range comprised between 12 and 13 months old).

Scrotal circumference was measured transversely, in correspondence with the greatest diameter of the scrotum, with a tape (Reliabull Scrotal Tape, Lane Manufacturing Inc., Denver, CO, USA) following the instructions of Hopkins et Spitzer [13].

Bulls classified as satisfactory for breeding were assigned to artificial insemination (AI), or to natural service (NS) categories, those classified while non-satisfactory were culled.

All the AI and NS bulls were examined (age: 424.6 ± 14.3 days old) in relation to their attitude to artificial vagina (AV) and for breeding soundness (BBSE) as outlined by the manual of the SFT (Society of Theriogenology) [4]. The prepuce, the penis, the scrotum, the testis, the epididymis, and the spermatic cord, as well as testicular consistency were evaluated. Furthermore, an assessment of the internal genitalia was performed via transrectal examination of the vesicular glands and ampullary ducts. 

### 2.2. Semen Analysis

One ejaculate was collected, from trained yearling bulls, mounting other bulls restrained in a short-sided breeding chute, by the artificial vagina (AV) method. If the first collection was considered quantitatively and/or qualitatively unsatisfactory, other two samplings were performed (collected 15 min apart). In case these latter did not reach the minimum threshold values for motility and concentration, bulls were considered as “deferred” and evaluated again after one month.

Semen quality was assessed as described by the SFT guidelines [4].

The volume (mL) of the ejaculates was measured using a 15 mL collection tube (code 62.554.502, Sarsted s.r.l., Verona, Italy) connected to AV. Total motility (defined as the percentage of spermatozoa that exhibit motility of any form) and progressive motility (defined as the percentage of spermatozoa that are moving in a rapid linear fashion) were estimated, visually, by an experienced veterinarian in charge of BBSE at the genetic center, as means of 10 microscopic fields at 37 °C using a phase-contrast microscope at 200× magnification (CH 2, Olympus Corporation, 2-3-1 Nishi-Shinjuku, Shinjuku-ku, Tokyo 163-0914, Japan) equipped with a heated stage (HT 50, Minitube, Minitüb GmbH. Hauptstraße 41 84184 Tiefenbach Germany). For the motility evaluation, raw semen was previously diluted to 1:60 in sterile 0.9% NaCl solution. Spermatozoa motility was evaluated after placing a 15 µL drop of semen on pre-warmed glass slide, covered by a 20 × 20 mm coverslip. The spermatozoa concentration was measured using a photometer (Accucell, IMV-Tecnologies, Rue Jean Moulin 78120, Rambouillet, France) by diluting 40 µL of semen in 3960 µL of 0.9% NaCl and the total number of spermatozoa was calculated by multiplying the ejaculate volume for sperm concentration. For the morphological evaluation of spermatozoa, a smear was prepared by mixing 7 µL of raw semen with 90 µL of eosin–nigrosin solution [32]. Morphological characteristics of spermatozoa (percentage of normal and abnormal spermatozoa) were assessed using bright-field microscopy, at 1000× magnification (Orthoplan, Leitz, Germany), by counting at least 200 sperm cells as defined by Barth and Oko [31]. Spermatozoa abnormalities were grouped into two categories: primary and secondary, as described by the SFT.

The starting dataset counted 740 bulls but, after a data editing, which discriminated both for missing information and for bulls that presented no satisfactory motility or morphology, a final number of 577 subjects were retained in the time interval from 2014 to 2021, and an age between 395 and 465 days. Of all the 577 involved bulls considered, 373 were satisfactory (64.6%) and 204 (35.4%) were deferred; however, these 204 deferred bulls were re-evaluated and at the end of the study, they reached the “satisfactory” score and were subsequently included in the dataset.

Table 1 reports the number of bulls evaluated for each month of the year in more detail.

To evaluate the effect of environmental temperature and humidity on bulls’ fertility, meteorological data were extrapolated from the meteorological station located at the airport area of Rivolto (Friuli Venezia Giulia) at 45°58′30.47″ N 13°02′56.68″ E, 30 km from the performance station. Based on the daily average temperature and humidity, THI was calculated for every day using the following formula [29]:THI = (0.8 × temperature) + ((humidity/100) × (temperature − 14.4)) + (46.4) 

To investigate the effect of environmental conditions, in short and medium time, on the quality of semen, the average THI in the previous 60, 30, and 10 days before the semen collection was calculated.

### 2.3. Statistical Analysis

Data for each bull and its ejaculates were recorded and imported into SAS software version 9.2 (SAS Institute Inc., 100 SAS Campus Drive, Cary, NC 27513-2414, USA) [33].

First, for semen quantity and quality data, descriptive analysis and correlations were made.

Then, the first analysis of variance tried to evaluate the impact of biological and management factors on semen traits, the model was fitted using the PROC-GLM of SAS software as follows:Y_ijklm_ = µ + D_i_ + A_j_ + W_k_ + S_l_ + e_ijklm_
where:
Y_ijklm_ = trait measured of the ejaculate collected;D_i_ = fixed effect of the date semen collection (i = 1…83);A_j_ = fixed effect of the age at semen collection (covariate);W_k_ = fixed effect weight at the end of the performance test (covariate);S_l_ = fixed effect of scrotal circumference at the end of the performance test (covariate);e_ijklm_ = random residual effect.

The second analysis of variance related semen features with the daily conditions of temperature and humidity during the period of study (summarized by calculating the TH index as described above).

Once again, the model was fitted using the PROC-GLM of SAS software as follows:Y_ijklm_ = µ + T_i_ +D_j(i)_ + A_k_ + S_l_ + e_ijklm_
where Y_ijklm_ = trait measured of the ejaculate collected;
T_i_ = fixed effect thi (i = 1…5; class 1th ≤ 40, class 2 40 < th ≤ 50, class 3 50 < th ≤ 60, class 4 60 < th ≤ 70, class 5th > 70);D_j(i)_ = fixed effect of the date semen collection (j = 1…83) within THI class;A_k_ = fixed effect of the age at semen collection (covariate);S_l_ = fixed effect of scrotal circumference at the end of the performance test (covariate);e_ijklm_ = random residual effect.

Bonferroni test was used to verify the significance of the differences among each value of the increasing classes of THI.

## 3. Results and Discussion

### 3.1. Descriptive Statistics

To summarize the information reported in the data set, descriptive statistics about fertility traits and the average daily gain are reported in Table 2.

The average daily gain for all included bulls was 1516.9 g/day with a maximum of 2052 g/day and minimum still higher than 1 kg gained per day, thus confirming the genetic selection oriented to dual-purpose for Simmental breed. When accounting for the year of study, the average gain constantly increased from 1330.5 g/d in 2014 to 1636.9 g/d in 2021, as shown in Figure 1.

The bulls of the study showed a mean testicular dimension of 37.4 cm, ranging from 30.0 to 44.0 cm. The SC was higher compared to the threshold expressed by the Society of Theriogenology, where the minimum average SC is equal to 30 cm for satisfactory bulls that are less than 15-month-old [4]. The mean semen volume was 3.3 mL with minimum and maximum varying from 1.0 to 8.0 mL, respectively, a very wide range of variation but relatively constant during the study (Table 1) and in accordance with that previously observed by other authors [34,35,36], and by Snoj et al. [25], who reported a mean semen value closer to the one of the present study, probably because it was carried out in Slovenia, that is a climate area similar to northeastern Italy.

Spermatozoa concentration showed high variability, ranging from 49.9 to 3005.0 × 10^6^ spz/mL, and a slightly lower average value than other reports [25,37,38]. No differences were evidenced throughout the years of collection. One possible explanation could be the young age of the bulls and the insufficient period for training to semen collection through an artificial vagina.

Despite the quite good average result for progressive motility (42.9%), in line with other observations [39], the annual trends showed a decreasing pattern with a minimum (34.7%) during 2018, and then a new rising pattern, reaching 39.8% in 2021 (Figure 2). However, all bulls evaluated in the case study were satisfactory because the average progressive motility was above the minimum 30% as recommended for at least a “Fair” classification, where motility has to be between 30% and 49% [40].

The spermatozoa morphological features showed high variability between the minimum and maximum total number of abnormalities (2.0 and 79.3%, respectively); however, they were far below the recommended limit of 30% [11] and no relevant differences were observed due to the year of sampling (data not shown). The results were in accordance with other studies, which also evaluated semen characteristics in bulls with similar age intervals (17–22 months and 14–19 months of age, respectively) [41,42].

Positive correlations were evidenced between the mean volume of ejaculate and the number of total spermatozoa (r = 0.66, *p* < 0.0001), and among spermatozoa concentration with the number of total spermatozoa (r = 0.80, *p* < 0.0001), total motility (r = 0.32, *p* < 0.0001), progressive motility (r = 0.32, *p* < 0.0001), and normal spermatozoa (r = 0.32, *p* < 0.0001). The percentage of normal spermatozoa was positively correlated with total and progressive motility (r = 0.44, *p* < 0.0001 and 0.43, *p* < 0.0001, respectively), and with the percentage of normal spermatozoa (r = 0.64, *p* < 0.0001). The positive correlations found between the number of alive sperms and total motility were in line with previous reports [24,43] and with Lasely [39] who found a similar correlation coefficient (0.46). Differently from other reports, which found strong negative relations between motile spermatozoa and both head and tail abnormalities [44], in this study, motility was only weakly correlated with the abnormal spermatozoa (r = −0.17 and r = −0.16, respectively, *p* < 0.0001). Negative correlations were evidenced also among the percentage of normal spermatozoa and abnormalities (data not reported).

### 3.2. Effects of Bull’s Age and Scrotal Circumference on Semen Parameters

Table 3 reports the results of the analysis of variance for the evaluation of the effect of the increase in bulls’ age on all the considered quantitative and qualitative semen traits. A positive effect, as evidenced by the calculated b coefficients, was observed between bull age and seminal characteristics, except for secondary abnormalities.

More in detail, the age at semen collection was statistically correlated with volume (*p* < 0.05), total motility (*p* < 0.01), progressive motility (*p* < 0.05), spermatozoa concentration (*p* < 0.01), the number of total spermatozoa (*p* < 0.01), the percentage of normal spermatozoa (*p* < 0.05), primary (*p* < 0.05), and total abnormalities (*p* < 0.05). For every extra day of age an increasing of the ejaculated volume (0.011 mL), of total and progressive motility (0.167% and 0.141%, respectively), of spermatozoa concentration (3.583 × 10^6^ spz/mL), of the number of spermatozoa in the entire ejaculate (0.018 × 10^9^), and of the normal sperm (0.081%) were produced; instead, the percentage of primary (−0.047%) and total abnormalities (−0.081) decreased. Those results confirm the general improvement of semen traits with the increasing age of bulls.

SC showed a significant association for spermatozoa concentration (*p* < 0.05) and number of total spermatozoa (*p* < 0.0001) while the other seminal parameters did not display any significant variation (NS). For every extra centimeter of SC, 17.5 × 10^6^ spz/mL and 0.102 × 10^9^ spermatozoa in the total ejaculated were produced. Similar results were also reported by Sylla et al. [27] who found a positive relationship between the scrotal circumference and semen concentration in three Italian beef breeds (Marchigiana, Romagnola and Chianina bulls), thus confirming that the dimensions of testis are, above all, representative of productive features. However, even if the results of our research were in accordance with former studies and the correlations had already been demonstrated in the literature for a long time, this is the first analysis conducted for the Italian Simmental breed.

### 3.3. Effects of the Environmental Conditions on Semen Parameters

When high environmental temperature and humidity overcome the possibility for the animals to maintain their physiological body temperature, the adaptive mechanisms fail, and heat-stress occurs [45]. These conditions also affect the testes and seminal attributes at any time of spermatogenesis (estimated to last around 60 days in bull) [46,47]. Thus, we chose to record THI for the previous 60, 30, and 10 days before the semen collection, in accordance with key moments of the spermatogenesis. The sixtieth day before the ejaculate collection is considered the moment of early spermatogenesis (cells are in the phase of spermatocytogenesis) while the thirtieth day corresponds to the average period of late spermatogenesis (cells are in the stages of spermiogenesis and meiosis). Finally, epidydimal sperm maturation takes around 10–11 days [32].

Table 4 summarizes the characteristics of the chosen five classes of THI (class 1 th ≤ 40, class 2 40 < th ≤ 50, class 3 50 < th ≤ 60, class 4 60 < th ≤ 70, class 5 th > 70) in terms of: number of observed bulls, temperature–humidity index, temperature, and humidity, considering the three moments of bull’s spermatogenesis.

The number of evaluated subjects varied from a minimum of 36 bulls to a maximum of 179 bulls. Overall, the environmental parameters were not too extreme. The THI ranged between 36.9 ± 2.3 and 73.7 ± 1.8, the temperature ranged between 1.4 ± 1.9 °C and 25.0 ± 1.4 °C, the humidity ranged from 68.5 ± 6% to 85.0 ± 16.5%.

To understand the way by which those semen characteristics were affected by the increasing THI, Figure 3 reports the corrected means for motilities (both total and progressive). While no significant correlations were reported at 60 days before the semen collection (NS), we observed a growing pattern for spermatozoa motility which was correlated with increasing THI classes at 30 and 10 days (*p* < 0.01 for both of them) before the semen collection (Table 5). More in detail, at 30 days before the semen collection, total and progressive motility increased from 39.3% to 56.4% and from 32.4% to 47.3%, respectively, and the values relative to THI at class 1 significantly differed with classes 3 (*p* < 0.01 and *p* < 0.05 for total and progressive motility, respectively), 4 and 5 (*p* < 0.01 for all the reported values). Similarly, total motility for the THI at class 2 differed also with classes 4 and 5 (*p* < 0.05 for both of them). Ten days before the semen collection, total and progressive motility was raised from 42.4% to 55.8% and from 34.8% to 55.6%, respectively. Only at class five was a mild decrease was reported, reaching the percentages of 55.2% and 45.6%. At this time point, the total motility, relative to the THI in class 1, was statistically different with classes 3, 4, and 5 (*p* < 0.05, *p* < 0.0001, and *p* < 0.01) while those relative to THI in class 2 differed with classes 4 (*p* < 0.01) and 5 (*p* < 0.05). Progressive motility for the THI in class 1 differed statistically with classes 4 and 5 (*p* < 0.0001 and *p* < 0.01, respectively), while the value of the THI for the class 2 differed with class 4 (*p* < 0.01).

Those results were surprising since a depression in motility was expected. However, also after an additional data editing, performed for excluding outlier bulls, the same grades of significance (*p* < 0.01) were maintained both at 30 and 10 days before the semen collection. This was probably due to the small number of outlier animals, which cannot invalidate the reliability of the statistical analysis. In this view, and to realistically show the differences in semen characteristics of young Italian Simmental bulls, those animals were not excluded from the analysis. Previous studies reported an overall depression for motility after a thermal-humidity insult. However, these experiments were performed through the method of the scrotal insulation [48,49,50,51]. Briefly, scrotal insulation exposes animals to extremely hot and humid conditions. In our study, no extreme THI intervals were created because the number of bulls included in an extreme class would have been too low to perform significant statistics. Moreover, differently from the scrotal insulation model, in our condition, thermal stress changes in the daytime with significant relief during the night. Furthermore, the weather conditions where the performance station is located are normally constant with a few extreme temperatures. The absence of an adverse effect on semen traits in a temperate climate was also reported by Llamas-Luceño et al. [28] who found no effect of the THI on fresh semen in young Holstein bulls. Thus, also in the case study, it was likely that animals could at least partially cope with thermal stress, except at 10 days before the semen collection, when a mild decrease in motility parameters for the THI class five could be observed.

A decreasing pattern for morphologically normal spermatozoa, correlated with the increasing THI, was reported at 60, 30, and 10 days before the semen collection (*p* < 0.0001 for all of them). Figure 4 reports the corrected means for the morphological semen traits. At 60 days before semen collection, the percentage of normal spermatozoa diminished (from 81.1% to 74.4%) and the percentages of primary (from 6.8% to 9.9%) and secondary (from 12.0% to15.8%) abnormalities increased. The percentage of morphologically normal spermatozoa for the THI at class 1 significantly differed with classes 4 (*p* < 0.05) and 5 (*p* < 0.01). The same appeared also for class 2, but with different levels of significance (*p* < 0.01 both for class 4 and 5). The percentage of primary abnormalities for the THI at classes 1 and 2 significantly differed with class 5 (*p* < 0.05 and *p* < 0.01, respectively). The percentage of secondary abnormalities for the THI at class 2 differed with classes 4 and 5 (*p* < 0.05 for both of them).

Similarly, 30 days before collection the percentage of normal spermatozoa passed from 80.0% to 74.2% and percentages of abnormalities ranged from 6.5% to 11.6% and from 13.4% to 16.0% for the primary and secondary ones, respectively. The percentage of morphologically normal spermatozoa for the THI at class 1 differed significantly with class 5 (*p* < 0.05). For the THI at class 2, the percentage of normal sperm differed with classes 4 and 5 (*p* < 0.0001 for both of them), while, for the THI at class 3, it differed significantly with class 5 (*p* < 0.01). Primary abnormalities for the THI in classes 1, 2, and 3 differed significantly with class 5 (*p* < 0.01, *p* < 0.0001, and *p* < 0.0001, respectively) and the THI in class 3 also differed with class 4 (*p* < 0.01). The percentage of secondary abnormalities differed significantly only among the THI at class 2 with classes 3, 4 and 5 (*p* < 0.0001, *p* < 0.01, and *p* < 0.0001). Moreover, at 10 days before the semen collection, the percentage of normal spermatozoa maintained a declining pattern, ranging from 82.7% to 70.4%. The percentages of abnormalities increased from 5.5% to 11.5% and from 11.8% to 18.1%, respectively for the primary and secondary ones. The percentage of morphologically normal spermatozoa, for the THI at class 1 reported a significant difference with classes 4 and 5 (*p* < 0.01 and *p* < 0.0001); similarly for the values of THI at classes 3, 4, and 5 reported a significant difference with class 5 (*p* < 0.0001, *p* < 0.01, and *p* < 0.01, respectively). The value for primary abnormalities for the THI at classes 1 and 2 differed significantly with classes 4 (*p* < 0.01 and *p* < 0.05) and 5 (*p* < 0.0001 for both). The percentage of primary abnormalities also differed for the THI at class 4 with class 5 (*p* < 0.05).

Finally, the percentage of secondary abnormalities for the THI at classes 1, 2, and 3 differed significantly with class 5 (*p* < 0.0001, *p* < 0.05, and *p* < 0.05, respectively).

The highlighted negative effects of growing THI during the initial phase of spermatogenesis have been reported by Sabés-Alsina et al. [29], at 60 days before the collection, and by Rhaman et al. [48] and Fernandes et al. [49], at 30 days before. Concerning the effect of weather conditions during the last 10 days before the collection, conflicting evidence has been reported [50]. By applying the scrotal insulation model, various authors did not report any effect when thermal stress was applied at time nearest to day of semen collection [48,49,51], while Kastelic et al. [52] found the greater percentage of midpiece and cytoplasmatic droplet defects for the spermatozoa, which were in the epididymis during the insulation, on days 5 and 8. The latter was similar to the case study, which reported a significantly increasing percentage of total sperm abnormalities, correlated with the increasing THI, 10 days before the semen collection (*p* < 0.0001).

The large number of sperm abnormalities observed here is not surprising and it is probably due to sexual immaturity as previously observed in other breeds [27,43,53,54,55]. The bulls were also experiencing their first attempts of semen collection by AV, and for this, some cases of incomplete ejaculation cannot be excluded. Bull handlers and semen collectors likely play a pivotal role [7,8] although investigating the effect of the operator at semen collection on seminal characteristics was beyond the aim of this work. However, the aforementioned motives could partly explain the unexpected trend of motility and morphology related to THI. The ejaculates considered in the study were the early samples collected from these young bulls and the germ cells were in the tail of the epididymis for a long time; as a consequence, the abnormalities tended to be higher, and probably overestimated, than those routinely collected from trained bulls, as proposed by Hirwa et al. [56] and Picard-Hagen et al. [57].

## 4. Conclusions

In this study, the variability of some seminal parameters on fresh semen in 577 Italian Simmental young bulls was evaluated. The results revealed that, for every extra centimeter of scrotal circumference, bulls produced a major number of spermatozoa per milliliter and a higher number of spermatozoa in the total ejaculated volume. Coherently with the physiological basis, almost all of the semen features improved significantly with the age of the bulls.

During the entire period of study, the percentage of normal spermatozoa was negatively affected by increasing THI. Unexpectedly, at 30 and 10 days before the semen collection, total and progressive motilities were positively influenced by the increasing THI. However, in our study, no extreme THI occurred, and the animals could at least partially cope with thermal stress, except for the THI at class five 10 days before semen collection, when a mild decrease could be observed. The study confirms that qualitative semen features are affected by climate variations in different ways, according to the moment in which the heat stress occurs.

## Figures and Tables

**Figure 1 animals-12-02476-f001:**
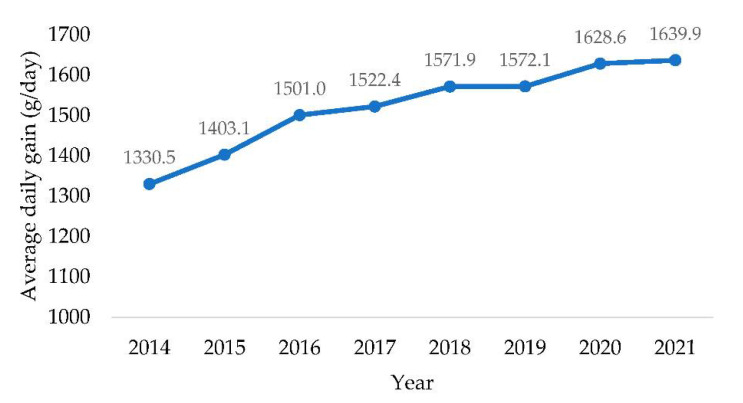
The trend of the average daily gains of 577 Italian Simmental young bulls recorded during performance test in the time interval comprised from 2014 to 2021.

**Figure 2 animals-12-02476-f002:**
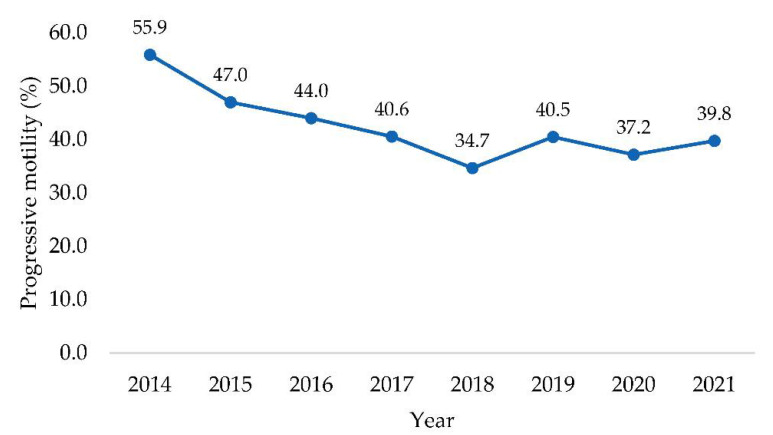
The trend of progressive motility of 577 Italian Simmental young bulls recorded in the time interval comprised from 2014 to 2021.

**Figure 3 animals-12-02476-f003:**
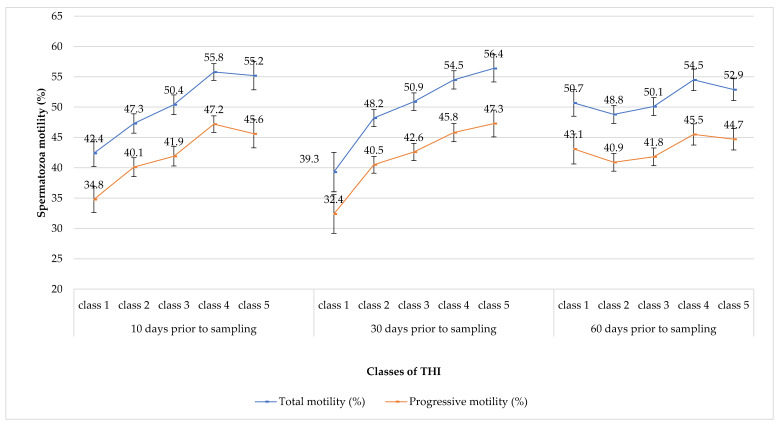
Trends of spermatozoa motility (both total and progressive) corrected means through the five increasing classes of THI for the three periods of spermatogenesis considered.

**Figure 4 animals-12-02476-f004:**
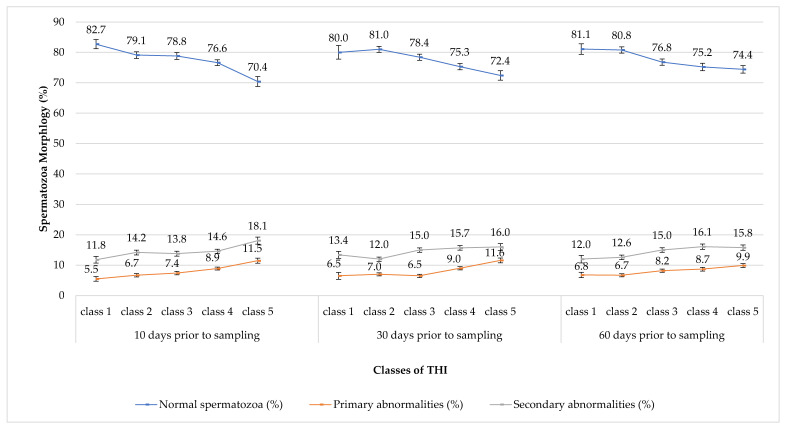
Trends of spermatozoa morphology (normal and abnormal sperm cells) corrected means through the five increasing classes of THI for the three periods of spermatogenesis considered.

**Table 1 animals-12-02476-t001:** Bulls evaluated per month.

Month of the Year	Obs
January	64
February	44
March	74
April	37
May	74
June	23
July	42
August	30
September	37
October	55
November	66
December	31
Total	577

Obs = number of observed bulls.

**Table 2 animals-12-02476-t002:** Descriptive statistics about physiological and seminal characteristics of Italian Simmental young bulls.

Observed Characters (*n* = 577)	Mean	SD	Min	Max	25% Q1	50% Med.	75% Q3
Age at semen collection (days)	424.6	14.3	395.0	465.0	416.0	426.0	437.0
Age at morphometry measure (days)	387.2	11.0	346.0	430.0	379.0	386.0	394.0
Average Daily Gain (g/day)	1516.9	173.8	1034.0	2052.0	1347.0	1457.0	1577.0
SC (cm)	37.4	3.3	30.0	44.0	35.0	37.5	39.5
Volume (mL)	3.3	1.1	1.0	8.0	2.5	3.2	4.0
Spermatozoa conc. (×10^6^/mL)	680.9	381.7	49.9	3005.0	404.6	613.7	885.6
TNS (×10^9^)	2.2	2.2	0.1	15.3	1.1	1.9	3.1
Total motility (%)	50.6	18.7	5.0	90.0	40.0	50.0	65.0
Progressive motility (%)	42.9	18.7	1.0	80.0	30.0	45.0	60.0
Normal Spermatozoa (%)	77.0	13.2	20.7	98.0	72.1	80.8	86.8
Primary abnormalities (%)	7.7	6.6	0.3	52.3	3.7	5.9	9.5
Secondary abnormalities (%)	14.3	9.9	1.0	71.7	7.8	12.3	19.2
Total abnormalities (%)	23.1	13.2	2.0	79.3	13.2	19.2	27.9

SC = scrotal circumference, TNS = total number of spermatozoa, 25% Q1 = interquartile1, Med.= Median, 75% Q3 = interquartile2.

**Table 3 animals-12-02476-t003:** Effect of the age at semen collection and scrotal circumference on seminal parameters.

Observed Characters (*n* = 577)	Total Deviance	Age at Semen Collection	SC	b Age at Semen Collection	b SC	r^2^
Volume (mL)	1007	*	NS	0.011	NS	20.6
Total motility (%)	201,782	**	NS	0.167	NS	33.2
Progressive motility (%)	191,101	*	NS	0.141	NS	31.6
Spermatozoa conc. (*10^6^/mL)	81,216,387	**	*	3.583	17.5	39.1
TNS (*10^9^)	2134	**	***	0.018	0.102	31.2
Normal spermatozoa (%)	97,780	*	NS	0.081	NS	33.7
Primary abnormalities (%)	25,602	*	NS	−0.047	NS	34.0
Secondary abnormalities (%)	56,306	NS	NS	NS	NS	40.9
Total abnormalities (%)	97,781	*	NS	−0.081	NS	33.7

NS = Non-Significant, *** =< 0.0001, ** =< 0.01, * =< 0.05. TNS = total number of spermatozoa, b = effect of increasing parameter (Age at semen collection, SC) on seminal characteristics, r^2^ = coefficient of correlation.

**Table 4 animals-12-02476-t004:** Characteristics of the classes (Mean ± SD) among the three different considered periods.

		Class 1	Class 2	Class 3	Class 4	Class 5
60 days before semen collection	**Obs**	53	158	146	118	102
	**THI**	38.3 ± 1.6	45.2 ± 2.8	55.1 ± 2.9	65.7 ± 2.9	73.7 ± 1.8
	**T (°C)**	1.6 ± 1.7	6.4 ± 2.1	12.6 ± 1.9	19.5 ± 1.9	25.0 ± 1.4
	**H (%)**	75.4 ± 16.0	80.0 ± 17.4	78.5 ± 15.7	72.7 ± 13.6	68.5 ± 6.6
30 days before semen collection	**Obs**	36	168	161	139	73
	**THI**	37.5 ± 1.1	44.8 ± 2.8	54.9 ± 2.6	65.4 ± 2.7	72.6 ± 2.2
	**T (°C)**	1.4 ± 1.9	6.4 ± 1.8	12.5 ± 1.7	19.3 ± 2.0	24.0 ± 1.5
	**H (%)**	78.8 ± 15.9	85.0 ± 16.5	74.8 ± 18.6	74.1 ± 11.0	73.5 ± 8.1
10 days before semen collection	**Obs**	77	128	129	179	64
	**THI**	36.9 ± 2.3	45.7 ± 2.4	53.5 ± 2.6	65.2 ± 2.8	73.5 ± 3.1
	**T (°C)**	1.5 ± 1.9	6.3 ± 2.7	11.5 ± 1.7	19.2 ± 1.9	24.4 ± 2.0
	**H (%)**	84.3 ± 13.4	75.6 ± 20.8	70.8 ± 12.6	74.5 ± 12,6	76.8 ± 6.2

Obs = number of observed bulls, THI = Temperature humidity index, T (°C) = Temperature and H (%) = Humidity.

**Table 5 animals-12-02476-t005:** Interaction between the THI, 60, 30, and 10 days before semen collection with seminal parameters, with level of significance.

	60 Days before Semen Collection		30 Days before Semen Collection		10 Days before Semen Collection	
	cl_th60	r^2^	cl_th30	r^2^	cl_th10	r^2^
Volume (mL)	**	20.8	NS	20.8	NS	20.8
TM (%)	NS	32.8	**	32.8	**	32.8
PM (%)	NS	31.2	**	31.2	**	31.2
Sperm. conc.(×10^6/mL^)	**	36.1	***	36.1	NS	36.1
TNS (×10^9^)	***	31.4	***	31.4	*	31.4
Normal spz (%)	***	33.6	***	33.6	***	33.6
Prim abn (%)	***	34.2	***	34.2	***	34.2
Sec abn (%)	***	40.4	**	40.4	***	40.4
Tot abn (%)	***	33.6	***	33.6	***	33.6

NS = Non-Significant, *** =< 0.0001, ** =< 0.01, * =< 0.05. TM= Total Motility (%), PM= Progressive Motility (%), Sperm. conc.= Spermatozoa concentration, TNS= total number of spermatozoa, Norm spz = Normal spermatozoa (%), Prim abn = Primary abnormalities (%), Sec abn = Secondary abnormalities (%) and Tot abn = Total abnormalities (%). cl_th60 = classes of THI of the last 60 days before semen collection, cl_th30 = classes of THI of the last 30 before days semen collection, cl_th10 = classes of THI of the last 10 days before semen collection, r^2^ = coefficient of correlation.

## Data Availability

Not applicable.

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
