# Peer review of "Environmental Factors Affecting the Reproductive Efficiency of Italian Simmental Young Bulls"

_animals, 2022, doi:10.3390/ani12182476_

Round 1

Reviewer 1 Report (New Reviewer)

Authors performed an interesting study in a local italian breed, revealing new data in a large sample size population. I would recommend them to improve the results section, in order to show results more clear and enhance clarity of the manuscript. I think there are some outlier bulls that interefere on the results comprehension. Maybe they could divide bulls into different groups to show results, or something like that as I comment when analysing total and progressive motility, for example. 

This is the major change I feel this manuscript needs. As well, I have some other comments:

- ln 16. explain BSE acronym

- ln 30. n=

- Why there are some track changes in the manuscript?

- ln 53. one of the most... In natural service there are some factors afecting fertility: nutrition, reproductive diseases, bull, etc.

- ln 54. BBSE: breeding bull soundness evaluation. Mention like this throughout the manuscript.

- ln 56. And even ultrasound and sanitary procedures, like in Spanish guide recently published.

- ln 58. managerial? Would you mean factos associated to handling or  management?

- ln 62. the first time you say an acronym, you must explain it. artificial insemination (AI). You must do this throughout the manuscript with the rest of acronyms.

- ln 70. managerial again.

- ln 71. delete "."

- ln 72. is not

- ln 76. summer

- ln 88. this study

- ln 89-90. the second most diffuse breed world-89 wide REFERENCE

- ln 87. missing connection from the introduction and introducing your study.

- ln 92-98. Objectives are clear, but you just mention in the introduction factors related to SC, and then you analyze some others... please complete the introduction giving an adequate state-of-the-art

- ln 109. I would include into brackets the italian region, indicating if north, south... more or less, just for readers to know instead of looking for the coordenates

- ln 110. delete and

- 2.1. Animals. I think you are over-explaining procedures out of the study. Please summarize and place the reference of these guidelines if somebody would like to know deeper.

- ln 147-152. How did you measure total and progressive motility? Subjectively? By CASA?

-ln 162-167. Reduce this length and say just the final number of bulls. 

-LN 170. near? How far? Specify

-ln 171. THI

- ln 179. version and complete data of SAS

- table 1. Did data follow a normal distribution? If yes, just say mean +-sd, avoid max/min. Ir it did not follow a normal distribution, express it as median and interquartile range

- table 1. results of total and progressive motility are not good at all. If you measure it in fresh semen and using artificial vagina, these values are extremely low... Express them differently, because we do see a very low minimums and high maxims... 

- ln 253. Sorry but in young bulls, fresh semen, and artificial vagina, having a 42.9% os progressive motility is not a good average result.

- ln 285. avoid saying fertility traits. Say semen quality or seminal parameters.

- table 4 is very interesting, but it includes all THI ranges. It would be very welcome to have, at least the same table for THI class 1 and class 5. This would give new and interesting information with the high sample size you provide

- figure 3 and 4 are completely necesary and very visual. Congratulations. I think that showing statistical differences would be positive. As well, include the error bars.

Author Response

[Au]: We want to point out that the lines indicated in the following document are referred to the highlighted version of the manuscript. Please see the attachment.

Authors performed an interesting study in a local italian breed, revealing new data in a large sample size population. I would recommend them to improve the results section, in order to show results more clear and enhance clarity of the manuscript. I think there are some outlier bulls that interefere on the results comprehension. Maybe they could divide bulls into different groups to show results, or something like that as I comment when analysing total and progressive motility, for example.

We tried to revise the way by which we performed the statistical analysis excluding the outlier bulls. However, both at 30 and 10 days before the semen collection the same grades of significance were maintained. This was probably due to the small number of outlier bulls, which could not invalidate the statistical analysis. For this reason and to realistically show the differences in semen characteristics of the Italian Simmental young bulls, they have not eliminated from the data set.

However, we add a couple of sentences in the “results and discussion section”. Changes are visible at lines 415-421.

The same type of analysis was performed also for morphological traits and no difference for the degree of significance were evidenced.

This is the major change I feel this manuscript needs. As well, I have some other comments:

- ln 16. explain BSE acronym

[Au]: We explain the BSE acronym and, following the suggestion for line 54, we changed the acronym in BBSE through the entire manuscript.

- ln 30. n=

[Au]: In accordance we changed it.

- Why there are some track changes in the manuscript?

[Au]: The present study is a resubmission of a previously rejected article. The editor encouraged us to resubmit it after extensive revisions, as a communication, and to highlight all the changes we made. We submitted both the “clean” and the “highlighted” version of the manuscript; unfortunately, the system sent to Reviewers the highlighted version. We are sorry for the inconvenience.

- ln 53. one of the most... In natural service there are some factors affecting fertility: nutrition, reproductive diseases, bull, etc.

[Au]: In accordance we partially rephrased the sentence because bull fertility is one of the most contributors to the herd reproductive success but not the only one. Change is visible at line 55.

- ln 54. BBSE: breeding bull soundness evaluation. Mention like this throughout the manuscript.

[Au]: We changed the acronym through the entire manuscript.

- ln 56. And even ultrasound and sanitary procedures, like in Spanish guide recently published.

[Au]: We added this information in the manuscript and we also cited the two papers from the VART guide in which the standard procedure for bull evaluation in Spain are described. The add sentence is visible at lines 60-61.

- ln 58. managerial? Would you mean factors associated to handling or management?

[Au]: In accordance we changed “managerial” with a more appropriate word. Change is visible at line 63.

- ln 62. the first time you say an acronym, you must explain it. artificial insemination (AI). You must do this throughout the manuscript with the rest of acronyms.

[Au]: In accordance we explained the acronyms when we cited them for the first time. We did it for all the acronyms throughout the manuscript hoping that now they are all well indicated.

- ln 70. managerial again.

[Au]: In accordance we changed “managerial” with a more appropriate word. Change is visible at line 82.

- ln 71. delete "."

- ln 72. is not

- ln 76. summer

- ln 88. this study

[Au]: We correct lines 71, 72, 76 and 88 as indicated.

- ln 89-90. the second most diffuse breed worldwide REFERENCE

[Au]: We decided that it was better to delete the sentence at all.

- ln 87. missing connection from the introduction and introducing your study.

[Au] We added new sentences in order to better introduce the aims of the work, accordingly to the Referee comment. We hope we successfully improved the manuscript. Changes are visible at lines 78-81.

- ln 92-98. Objectives are clear, but you just mention in the introduction factors related to SC, and then you analyse some others... please complete the introduction giving an adequate state-of-the-art

[Au]: We completed the introduction to better explain the state-of-art. The added sentence is visible at lines 104-110.

- ln 109. I would include into brackets the italian region, indicating if north, south... more or less, just for readers to know instead of looking for the coordenates

[Au]: We agree with the suggestion and, near the coordinates, we added also the region and the area where the study was performed. The adding sentence is visible at lines 134-135.

- ln 110. delete and

[Au] We delete it.

- 2.1. Animals. I think you are over-explaining procedures out of the study. Please summarize and place the reference of these guidelines if somebody would like to know deeper.

[Au]: We tried to summarize the description about the animals involved in the study. The reference for the manual of the Society of Theriogenology is indicated with the number 4 at line 163.

- ln 147-152. How did you measure total and progressive motility? Subjectively? By CASA?

[Au]: Total and progressive motility were evaluated always by the same experienced veterinarian in charge of BBSE at the genetic centre, with 30 years of experience. We better specified it at line 180.

-ln 162-167. Reduce this length and say just the final number of bulls.

[Au]: in Accordance we shortened the sentence indicating only the final number of evaluated bulls.

-LN 170. near? How far? Specify

[Au]: We specify the distance between the airport area of Rivolto and the location of the performance station. The change is visible at lines 215.

-ln 171. THI

[Au]: We changed TH index only with the acronym THI.

- ln 179. version and complete data of SAS

[Au]: We added the version of SAS used for the statistical analyses of the manuscript at line 225

- table 1. Did data follow a normal distribution? If yes, just say mean +-sd, avoid max/min. Ir it did not follow a normal distribution, express it as median and interquartile range

[Au]: Yes, data follow a normal distribution but for a major explanation we add to table 2 also the median and interquartile range in which we show that the larger part of the observations were in the central interquartile.

- table 1. results of total and progressive motility are not good at all. If you measure it in fresh semen and using artificial vagina, these values are extremely low... Express them differently, because we do see a very low minimums and high maxims...

[Au]: We are conscious about the low motility values obtained but, as outlined for the large number of sperm abnormalities observed, this is not uncommon and it is probably due to:

- sexual immaturity

- the fact that bulls are yearling and for this experiencing their first attempts of semen collection by AV and the spermatozoa were in the tail of the epididymis from a lot of time.

In other words, these results cannot be compared to those of bull where seminal production was “stabilized” by a routine seminal collection (as in artificial insemination centres).

Our evaluation of progressive motility is also performed in diluted samples allowing individual cells motility evaluation, where each non-progressive type of motility is not considered as progressive. It is relevant to this point that in the last version of the SFT manual for BBSE (2018), for the evaluation of progressive motility, an adequate dilution of the sample is recommended (page 85).

- ln 253. Sorry but in young bulls, fresh semen, and artificial vagina, having a 42.9% os progressive motility is not a good average result.

[Au]: We have already answer to this aspect in the above comment.

- ln 285. avoid saying fertility traits. Say semen quality or seminal parameters.

[Au]: We changed “fertility traits” with “seminal parameters” through the manuscript.

- table 4 is very interesting, but it includes all THI ranges. It would be very welcome to have, at least the same table for THI class 1 and class 5. This would give new and interesting information with the high sample size you provide.

[Au]: We already performed the analysis to evidence the significant differences among all the different classes of THI by using the Bonferroni test. We are sorry but at this point we do not understand the request of the reviewer.

- figure 3 and 4 are completely necesary and very visual. Congratulations. I think that showing statistical differences would be positive. As well, include the error bars.

[Au]: We thank the reviewer about his positive feedback regarding figures 3 and 4.

Considering the statistical differences among all the different classes of THI, in the beginning we also considered the possibility to add them to the figures but then we prefer to avoid it because of the difficulties to express them clearly. Since almost all of the classes differed from the others, it is almost impossible to express those degrees of difference in a visual way without making the figures heavy to read.

Considering the error bars, we add them to the figures 3 and 4.

Reviewer 2 Report (New Reviewer)

This study shows the important of the confluence of various factors that determine the reproductive success of Italian simmental young bulls from inner factors like scrotal circumference to the average temperature and humidity as environmental conditions.

Lines 21, 152, 248, 258, 265, 301, 342, 352: Remove the letter L to the word spermatozoal.

Line 62: Author must specify the complete concept referred to by the abbreviation AI industry (artificial insemination).

Line 71: remove the period after the word “and” and continue the sentence in lowercase.

Line 87: remove the period after the word “characteristics”.

Lines 267,268: Remove the word live to Normal live sperm.

Line 355 and figure 3 (title and graph): remove motilities (motility).

Table 1: write correctly the term “total motility”

Author Response

Au]: We want to point out that the lines indicated in the following document are referred to the highlighted version of the manuscript. Please see the attachment.

This study shows the importance of the confluence of various factors that determine the reproductive success of Italian Simmental young bulls from inner factors like scrotal circumference to the average temperature and humidity as environmental conditions.

[Au]: We thank the reviewer for his positive feedback regarding the manuscript and for his correct interpretation about the aims of the manuscript.

Lines 21, 152, 248, 258, 265, 301, 342, 352: Remove the letter L to the word spermatozoal.

 [Au]: We remove the letter L from the word spermatozoal through the manuscript.

Line 62: Author must specify the complete concept referred to by the abbreviation AI industry (artificial insemination).

 [Au]: In accordance we explained the acronym when we cited it for the first time. We did it for all the acronyms throughout the manuscript hoping that now they are all well indicated.

Line 71: remove the period after the word “and” and continue the sentence in lowercase.

 [Au]: We rephrased the period after the word “and”.

Line 87: remove the period after the word “characteristics”.

 [Au]: We remove the period after the word “characteristics”.

Lines 267,268: Remove the word live to Normal live sperm.

 [Au]: We remove the word “live” in the two indicated lines.

Line 355 and figure 3 (title and graph): remove motilities (motility).

 [Au]: We correct the word through the manuscript.

Table 1: write correctly the term “total motility”

 [Au]: We correct the word and enumerate the table as Table 2 because of the addition of a new table as Table 1.

Reviewer 3 Report (New Reviewer)

Comments to the Authors

This study is an attempt to explain the “Environmental factors affecting the reproductive efficiency of 2 Italian Simmental young bulls”. It is not clear to me the purpose of the experiment as it does not have treatments or breeding values to compare or at least it is not expressed in the manuscript. The data presented in the manuscript are not novel. Therefore, it is not clear what exactly the authors are evaluating in the current manuscript. The data present several caveats and the results are biased as the bulls are not coming from the same source (breeder) and they do not have similar genetics. The Introduction does not lead to the objectives of the manuscript as it does not report the impact of environmental conditions on the fertility trait tested. Moreover, in Mediterranean conditions, the temperature is quite stable. To me, it is not a scientific manuscript; it is a Simmental breeder’s association report. The manuscript is rejected.

I listed my other concerns below in the order I found them in the manuscript.

Simple summary

-       L16. Please define “BSE”

-        

Abstract

The abstract needs a couple of sentences explaining the M&M of the experiment, especially regarding nutritional management and collection methodology.

-       L31-L41. Results are hard to follow since no M&M was explained before.

-       L32-L34. Please provide the age range.

-       L34-L36. Please provide the THI range.

-       L92-L97. What is the novelty of the experiment?

Introduction

Authors need to revise the final version of the manuscript before submitting it as the manuscript presents “changes” that were not “accepted”. The Introduction does not lead the reader to the objectives of the experiment. With the information provided in the Introduction is hard to see the novelty of the experiment. Moreover, historical environmental data from the experimental/location/state site is needed to interpret whether is a constant condition or just in specific years.

-       L52-L53. Not necessarily, female nutritive and reproductive status play an important role in fertility rates.

-       L59-L60. Wider knowledge? Like what? What has not been published on the parameters mentioned before?

-       L60-L61. Perhaps this is true for young bulls, however, they must reach sexual maturity first which is manipulated by extrinsic and intrinsic factors (Kenny and Byrne 2018; animal 12:S1).

-       L65-L69. Thus, which is the novelty of the current experiment?

-       L70-L73. Is this sentence related to young or mature bulls? The previous paragraph was about young bulls. I consider that the authors need to provide a couple of lines explaining the duration of the spermatogenesis and how this is affected by extrinsic factors.

-       L75-76. Please provide the breed and age of the bulls used in the experiment. Please provide the average temperature reported in the experiment conducted by Snoj et al.

-       L76-L79. Similar to before, more information regarding the experiment Bhakat is necessary.

-        

Material and Methods

-       L103-L106. How many animals were used in the experiment? Were born in the same year?

-       L105-L107. So, the bulls were assessed in the same performance station; however, they were born in different places, and more importantly, the genetic background differs among bulls. Am I correct?

-       L112-114. Please provide more details regarding the diet provided during the experiment. How much protein and energy were contained in the diet?

-       L122-L123. Please provide the age range.

-       L127-L128. Were the bulls used in the experiment classified as satisfactory?

-       L137-L142. Please be more precise. When was the ejaculate collected? How many bulls were deferred? How many samples per bull/year? This is important as the authors are associating the environmental conditions with the semen quality.

-       L162-L163. Importantly, as I mentioned previously, authors need to provide historical environmental conditions to understand the results of the experiment. In addition; as the source of the bulls differed, it is important to mention if the bulls were sourced from a different country or any place with different environmental conditions.

-       L168-172. Please be more specific. Were the meteorological data collected daily/weekly/monthly? Which month was fitted in the statistical analysis?

-       L174-L176. A table is needed indicating how many Bulls were evaluated per month and indicate the month of evaluation.

-       L179-L213. The statistical analysis presents a big caveat. I understand that the authors considered each bull as one experimental unit; however, bulls were not randomly combined in groups of 5-6. In the statistical analysis, each group should have been considered as one experimental unit.

-       In addition, the year needs to be added as co-variables in the statistical analysis as not all the bulls were assessed all the years. The maximal and minimal temperature should have been added as covariates in the statistical analysis.

Results

The correlations are interesting; however, are not novel. Why authors did not measure internal temperature at the moment of semen collection?

-       L224-228. These data are hard to follow due to the missing nutrition diet during the experiment and since bulls have a different genetic background (they are not coming from the same source).

-       L291-L300. What about the association with body weight?

-       L313-L322. Interesting; however, why the authors did not measure the internal temperature at the moment of semen collection? This could have indicated whether the animal presented heat stress or not.

-       LL324-L327. It is not clear when this temperature was recorded. Was the daily average for a specific month or year?

-       L349-L366. To have an accurate interpretation, the authors should have included in the statistical analysis the body weight or the daily live weight gain of the bulls. As the authors mentioned before, the average daily live weight gain ranged between 1500 to 2050 g per day. This indicates that the genetic background was not similar.

Conclusion

No comments.

Tables

-       Please provide the number of observations for each variable tested.

-        

Author Response

[Au]: We want to point out that the lines indicated in the following document are referred to the highlighted version of the manuscript. Please see the attachment.

This study is an attempt to explain the “Environmental factors affecting the reproductive efficiency of 2 Italian Simmental young bulls”. It is not clear to me the purpose of the experiment as it does not have treatments or breeding values to compare or at least it is not expressed in the manuscript. The data presented in the manuscript are not novel. Therefore, it is not clear what exactly the authors are evaluating in the current manuscript. The data present several caveats and the results are biased as the bulls are not coming from the same source (breeder) and they do not have similar genetics. The Introduction does not lead to the objectives of the manuscript as it does not report the impact of environmental conditions on the fertility trait tested. Moreover, in Mediterranean conditions, the temperature is quite stable. To me, it is not a scientific manuscript; it is a Simmental breeder’s association report. The manuscript is rejected.

 [Au] We thank the referee for pointing out the weak parts of the manuscript, however there are some points where we completely disagree from his point of view.

First of all, the manuscript is related to field study on yearling bulls participating to a growth performance test. Both historic and also recent researchs related to the definition of reproductive characteristic of bulls have the same approach (Awda, B. J.; Miller, S. P; Montanholi, Y. R; Vander Voort G., Caldwell, T., Buhr, M. M, Swanson, K. C. The relationship between feed efficiency traits and fertility in young beef bulls. Can. J. Anim, Sci. 2013, 93, 185-192. Fontura, A. B. P.; Montanholi, Y. R.; Diel de Amorim, M.; Foster, R. A.; Chenier, T.; Miller, S. P. Associations between feed efficiency, sexual maturity and fertility-related measures in young beef bulls. Animal. 2016, 10, 96-105).

The reasons for this kind of research are double: 1) setting out reference values for researchers, practitioners and breeders, 2) setting out reference values for a specific breed in a specific breeding situation. In this respect, there were no data relative to Italian Simmental yearling bulls already published. The existing bibliographic data relative to this breed is mainly related to animals different for age, breeding condition and country of origin. In other words, our data set is related to a group of animals, uniform, not only for breed and age, but also for environmental, nutrition and management conditions. In the era of genomic selection, the finding of phenotypic traits that can be associated with genomic differences is highly required, particularly for reproductive traits that are usually not easily heritable.

Relatively to the source of animals, obviously they not came from the same breeder, not belonging to laboratory animal species. Worldwide, in fact, bull calves intended for future semen production and genetic improvement always come from different breeders. In our case, male calves enter into the genetic centre at 20 to 40 days of age; after that they are raised in the same condition (feeding, weather, handling…) and differences in daily average gain, muscle/fat and so on, could be attributed to the genetics of each bull. Thus, desired traits could be selected. We also would like to remember that inbreeding is utilized in cattle breeding plans for selecting desired traits, but attention is also paid to avoid excessive inbreeding, which could lead to unwanted genetic defects.

I listed my other concerns below in the order I found them in the manuscript.

Simple summary

 -          L16. Please define “BSE”

[Au]: Accordingly, we define the acronym BSE and we changed it in BBSE (throughout the manuscript) to a better explanation.

Abstract

The abstract needs a couple of sentences explaining the M&M of the experiment, especially regarding nutritional management and collection methodology.

-     L31-L41. Results are hard to follow since no M&M was explained before.

[Au]: The abstract has a limited number of words (200) to be accepted; for this reason, we cannot describe M&M in detail. We added the sentence (visible at lines 31-32) regarding the method of semen collection but, due to several results that have to be reported regarding the effect of different factors on semen characteristics, we preferred not to focus on the nutritional information.

-           L32-L34. Please provide the age range.

[Au]: We provide the age range of the bulls involved in the study at line 36.

-           L34-L36. Please provide the THI range.

[Au]: We provide the THI range considered for the study at line 37.

-           L92-L97. What is the novelty of the experiment?

 [Au]: No reports regarding the association between the scrotal circumference ad semen characteristics are available for Italian Simmental bulls. In addition, the larger amount of the studies concerning the evaluation of the long-term effects of temperature and humidity on seminal characteristics are performed by the scrotal insulation model while only few reports evaluate the interaction between those traits and the THI, in cattle, independently on the breed. In compliance, the evaluation of the heat stress manifestation also in a normally temperate area (like the north-east of Italy) could be useful to point out the effects of the increasing global temperature scenario also on farm animals, which represent a great factor of concern nowadays.

Introduction

Authors need to revise the final version of the manuscript before submitting it as the manuscript presents “changes” that were not “accepted”. The Introduction does not lead the reader to the objectives of the experiment. With the information provided in the Introduction is hard to see the novelty of the experiment. Moreover, historical environmental data from the experimental/location/state site is needed to interpret whether is a constant condition or just in specific years.

[Au]: We submitted both the “clean” and the “highlighted” version of the manuscript; unfortunately, the system sent to Reviewers the highlighted version. We are sorry for the inconvenience. We also modified the introduction in order to improve the link between the state of the art and the aim of the work (changes are visible at lines 78-81). In addition, we added a sentence to better specify the region where the performance station is located and also the distance from the airport area where weather conditions are recorded (changes visible at lines 134-135 and 215).

-     L52-L53. Not necessarily, female nutritive and reproductive status play an important role in fertility rates.

[Au]: In accordance we partially rephrased the sentence because bull fertility is one of the most contributors to the herd reproductive success but not the only one. Change is visible at line 55.

  • L59-L60. Wider knowledge? Like what? What has not been published on the parameters mentioned before?

[Au]: Studies which considered environmental conditions often evaluated only the effect on the day of semen collection, ignoring the long-term consequences of extreme weather on the entire spermatogenic cycle and thus on seminal characteristics. To demonstrate this hypothesis, we considered the effects of temperature and humidity on seminal characteristics also on the day of semen collection but without significant results. Other studies, which also considered long-term interactions, only used temperature. The THI, which combines temperature and humidity, has been recently introduced as a better indicator of thermal stress in cattle because of its objectively-described threshold levels.

-        L60-L61. Perhaps this is true for young bulls, however, they must reach sexual maturity first which is manipulated by extrinsic and intrinsic factors (Kenny and Byrne 2018; animal 12:S1).

[Au] We agree with the observation of the Referee and accordingly, we added his/her suggestion into the text at lines 68-70.

-        L65-L69. Thus, which is the novelty of the current experiment?

[Au]: in order to better highlight the novelty of the results reported herein, we improved the introduction section by specifying that previous research focused mainly on the short-term effect of temperature on seminal characteristics, while ignoring THI. Please, see lines 104-110 in the introduction.

-        L70-L73. Is this sentence related to young or mature bulls? The previous paragraph was about young bulls. I consider that the authors need to provide a couple of lines explaining the duration of the spermatogenesis and how this is affected by extrinsic factors.

[Au]: in accordance we partially rephrased the sentence, changes are visible at lines 83-84.

-     L75-76. Please provide the breed and age of the bulls used in the experiment. Please provide the average temperature reported in the experiment conducted by Snoj et al.

[Au]: In accordance we specify the breed, the age and also the average temperatures in the study performed by Snoj et al. Changes are visible at lines 89-91.

-     L76-L79. Similar to before, more information regarding the experiment Bhakat is necessary.

[Au]: In accordance, more information regarding the average temperatures and also the age of the bulls are specified. Changes are visible at lines 96-98.

Material and Methods

-      L103-L106. How many animals were used in the experiment? Were born in the same year?

[Au]: The final number of bulls used for the experiment was 577. Since the study was performed from 2014 until 2021 the involved bulls were not born in the same year.

-        L105-L107. So, the bulls were assessed in the same performance station; however, they were born in different places, and more importantly, the genetic background differs among bulls. Am I correct?

[Au]: Yes, worldwide, bull calves intended for semen production and genetic improvement always come from different places/breeders. However, they enter the genetic station early in life (at 20 to 40 days of age in our case) and then they are raised under the same feeding, housing, handling (…) conditions. Thus, differences for example in daily weight gain of each bull are attributed to its own genetics and desired traits could be selected.

-        L112-114. Please provide more details regarding the diet provided during the experiment. How much protein and energy were contained in the diet?

[Au]: We add some more sentences related to the diet, changes are visible at lines 143-147.

-        L122-L123. Please provide the age range.

[Au]: The morphological evaluation of bulls is performed between 12 and 13 months old. The add sentence is visible at line 155.

L127-L128. Were the bulls used in the experiment classified as satisfactory?

L137-L142. Please be more precise. When was the ejaculate collected? How many bulls were deferred? How many samples per bull/year? This is important as the authors are associating the environmental conditions with the semen quality.

[Au]: We answer to these two comments in the same section.

In order to clarify the dataset dimension, we amended the paragraph 2.2, lines 195-206: The starting dataset counted 740 bulls but, after a data editing, which discriminate both for missing information and for bulls that presented no satisfactory motility or morphology, a final number of 577 subjects were retained in the time interval from 2014 until 2021, and an age between 395 and 465 days. Considered all the 577 involved bulls, 373 were satisfactory (64.6%) and 204 (35.4%) were deferred; however, these 204 deferred bulls were re-evaluated and at the end of the study, they reached the “satisfactory” score and were subsequently included into the dataset. Moreover, the ejaculates are collected every month (1 or 2 days of collection are performed on a random day of each month) and in order to include only one observation for each bull, only the last visit has been included for the statistical analysis.

-      L162-L163. Importantly, as I mentioned previously, authors need to provide historical environmental conditions to understand the results of the experiment. In addition; as the source of the bulls differed, it is important to mention if the bulls were sourced from a different country or any place with different environmental conditions.

[Au]: The initial sources of bulls are different but, when subjected to semen collection, they were at the performance station for a long period. Young bull calves entered the genetic centre when they were 25 ± 10 days and are evaluated for semen characteristics at the average age of 424.6 ±14.3, as described in lines 137 and 161. During all this period they are bred in the same environmental and nutritional conditions because of the importance to maintain them stable during the period of the growth performance test. No one of the bulls came from a different Country, all of them came from regions in the North-East of Italy; so, weather conditions didn’t change. Anyway, as mentioned before, calves enter to performance station when they are so young and are adapted to the environment for months before semen collection.

-      L168-172. Please be more specific. Were the meteorological data collected daily/weekly/monthly? Which month was fitted in the statistical analysis?

[Au]: The meteorological data were collected daily and derived from the airport area of Rivolto, located 30 km from the performance station. So, it is the daily average temperatures and humidity collected for all the months of the year (for all the years of study, from 2014 to 2021). The sentence is already present at line 216.

. No one of the considered months were fitted because we considered the date at semen collection. The statistical model takes into account the date at semen collection which includes also the THI, as nested, into the date itself.

-        L174-L176. A table is needed indicating how many Bulls were evaluated per month and indicate the month of evaluation.

[Au]­: In accordance we add a new table to better evidenced how many bulls were evaluated for each. We named the table as Table 1 and updated the other tables enumeration.

-        L179-L213. The statistical analysis presents a big caveat. I understand that the authors considered each bull as one experimental unit; however, bulls were not randomly combined in groups of 5-6. In the statistical analysis, each group should have been considered as one experimental unit.

[Au]: Actually, young bulls are raised in 5-6-unit groups until 12 months of age. Then, morphologically satisfactory bulls are moved to individual boxes. Only at this point they undergo semen collection and analysis. We agree with the referee that this method must be elucidated in the manuscript, thus we added this information in lines 148-149.
Based on this, we believe that the experimental unit is represented by the individual bull.

-        In addition, the year needs to be added as co-variables in the statistical analysis as not all the bulls were assessed all the years. The maximal and minimal temperature should have been added as covariates in the statistical analysis.

[Au]: The year of semen collection was not considered because we decided to include in the statistical model the date at the semen collection, which permits to better correct the data instead of considering only the year.

In addition, worldwide, if possible, genetic evaluation models for productive traits (for instance in cows) always use the environmental conditions of the specific day of data collection; thus allowing to obtain more cleaned data in comparison with the correction only for the year.

For a deeper explanation, we report the following example: In the case study, considering in the model only the effect of the year, the calculated r2 for total motility was equal to 5% while, considering the date at the semen collection, the calculated r2 was equal to 33%.

Results

The correlations are interesting; however, are not novel. Why authors did not measure internal temperature at the moment of semen collection?

[Au]: it should be considered that semen collection was performed under the direction of a veterinarian and that all bulls were submitted to clinical examination, and that animals which underwent heat stress would have been noted. Finally, due to the absence of extreme THI classes, severe heat stress which animals were not able to cope with (thus showing tachypnea, tachycardia, reduced feed intake, open mouth-breathing ecc) were not observed.

-        L224-228. These data are hard to follow due to the missing nutrition diet during the experiment and since bulls have a different genetic background (they are not coming from the same source).

[Au]: We already answered to this comment in the above comments.

-        L291-L300. What about the association with body weight?

[Au]: During the statistical analysis, by the ANOVA test, we tried to relate the last body weight available measure with semen features. However, the obtained results were no-significant for all the semen traits involved. So, to present only the significant data in a more visual way we decided to not to add those results in table 2, where only the associations among semen traits with both bull’s age and scrotal circumference are evidenced.

  • L313-L322. Interesting; however, why the authors did not measure the internal temperature at the moment of semen collection? This could have indicated whether the animal presented heat stress or not.

[Au]: See the comment above related to the first sentence regarding results.

-        LL324-L327. It is not clear when this temperature was recorded. Was the daily average for a specific month or year?

[Au]: See the comment above.

  • L349-L366. To have an accurate interpretation, the authors should have included in the statistical analysis the body weight or the daily live weight gain of the bulls. As the authors mentioned before, the average daily live weight gain ranged between 1500 to 2050 g per day. This indicates that the genetic background was not similar.

[Au]: As mentioned before, we tried to include the last available body weight in the statistical analyses but the results were no-significant for all the different semen characteristics involved in the study. All the bulls that arrive at the performance station are submitted to a performance test to evaluate their meat production attitude because bulls’ selection is oriented only on dual purpose production. So, since the selection is oriented also for meat production, an increase in the average daily gain through the years of study (from 2014 to 2021) is a normal and desirable result.

Conclusion

No comments.

Tables

-           Please provide the number of observations for each variable tested.

[Au]: The number of observations available for each variable tested are equal to 577. The starting data set was edited, and animals with any missing data were excluded.

Round 2

Reviewer 1 Report (New Reviewer)

Authors accomplished all the requierements that I previously did. I think the manuscript has been positively improved. 

I still surprised because of th e low seminal quality of these bulls, but it has been stated in the manuscript. 

I congratulate their work and effort. 

Reviewer 3 Report (New Reviewer)

The manuscript has been considerably improved

This manuscript is a resubmission of an earlier submission. The following is a list of the peer review reports and author responses from that submission.

Round 1

Reviewer 1 Report

In the study presented the authors tried to correlate some fertility traits of young bulls of the Simental breed with "climatic variations" by "short-time" or "long-term" effects of THI (temperature humidity index) during the previous 10, 30 and 60 days before semen collection. I believe the expression "climatic variation" is wrong in this context since climate is defined as the summary of weather over a period of 30 years. I would propose to use "weather changes".

15: Bull subfertility is a great contributor of the herd reproductive failure - Is that really true? Breeding bulls are raised in special breeding institutions where they are tested beside other traits for fertility...

19: The relationsship between scrotal circumference and semen characteristics is well known and was published long time ago (as review see: Understanding and evaluating bovine testes John P. Kastelic Theriogenology 81 (2014) 18-23. The influence of bovine breed on these results seems to be insignificant since similar results have been obtained for other domestic species. Therefore, the results published here are not novel, more in accordance with the former results.

Concerning the results in relation with temperature humidity index (THI), they should be discussed in more detail with those obtained by (16) Llamas-Luceño et al. High temperature-humidity index compromises sperm quality and fertility of Holstein bulls in temperate climates J. Dairy Sci. 103:9502–9514 https://doi.org/10.3168/jds.2019-18089. Interestingly these authors got only THI dependent differences during cryopreservation of bull semen.

In my opinion, the discussion should more focus on the results of THI and the different intervals chosen.

Author Response

Dear editor we corrected the manuscript addressing the reviewers' comments point-by-point and explaining the details of the revisions in our responses to their comments.

We hope that manuscript could be acceptable now for publication.

Below there are point-by-point responses to reviewer.

Reviewer # 1

In the study presented the authors tried to correlate some fertility traits of young bulls of the Simental breed with "climatic variations" by "short-time" or "long-term" effects of THI (temperature humidity index) during the previous 10, 30 and 60 days before semen collection. I believe the expression "climatic variation" is wrong in this context since climate is defined as the summary of weather over a period of 30 years. I would propose to use "weather changes".

[Au]

We first of all thank the Referee for taking the time to review this manuscript and for his/her comments and suggestions. We agree with changing the term as suggested above. We replaced with “weather changes” trough the manuscript. Concerning other comments, we provided a point-by-point list of answers as follows.

15: Bull subfertility is a great contributor of the herd reproductive failure - Is that really true? Breeding bulls are raised in special breeding institutions where they are tested beside other traits for fertility...

[Au]

According to your observation, we improved the sentence.

19: The relationship between scrotal circumference and semen characteristics is well known and was published long time ago (as review see: Understanding and evaluating bovine testes John P. Kastelic Theriogenology 81 (2014) 18-23. The influence of bovine breed on these results seems to be insignificant since similar results have been obtained for other domestic species. Therefore, the results published here are not novel, more in accordance with the former results.

[Au]

We agree with your observation. Thus, we proceeded with shortening the discussion section when accounting for the effect of scrotal circumference. Changes are visible in lines from 416-426.

Concerning the results in relation with temperature humidity index (THI), they should be discussed in more detail with those obtained by (16) Llamas-Luceño et al. High temperature-humidity index compromises sperm quality and fertility of Holstein bulls in temperate climates J. Dairy Sci. 103:9502–9514https://doi.org/10.3168/jds.2019-18089. Interestingly these authors got only THI dependent differences during cryopreservation of bull semen.

In my opinion, the discussion should more focus on the results of THI and the different intervals chosen.

[Au]

We agree about the necessity of a more focus on the results of THI in particular in comparison to the results reported by Llamas-Luceño et al., who found no influence of THI in a temperate climate. In this perspective, we added lines from 483 to 489 in the discussion section.

Reviewer 2 Report

This is a scientific domains were much work has been published. By example, see the references of this manuscript. Discussion is too big . It seams a bibliographic review. Many items were not treated in this manuscript.  4 pages is too much. This is not a a review. Also conclusion needs to be shortened. This manuscript has many contradictory and unexpected results, namely of THI on progressive motility. Only in THI 5, there is a mild decrease. In this work, nothing is written concerning   figures 2 and 3. if the several results are different (P<0.05.. I think that many values are not different (P<0.05) and the authors  seems to indicate  that they are different . I am somewhat confuse. Also some results are not published in tables (lines 249-251; 465-467). .Line 506, Where are the results on the day of semen collection?; Mass motility-How it was evaluated; for me mass motility is evaluated in a scale 0-5?;  Lines 490-491- Any temperature was taken from the animals (see also lines, 490-491??) ?; this is somewhat unexpected?. Where are these results of correlation (line3 466-467)?  Confuse lines 469-470?;  Unexpected results (lines 546-548)?; this manuscript hava many contradictory results. Also we see in discussion different methods, scrotal insulation (other manuscripts) vs present manuscript  (see lines 548-549); 

Line 498, I don´t agree. So many publications in male;  Many speculative  , see lines 557-558 and 573;  What means panting (line 494);  Some ortographic errors (lines 339,  537; 555; ; change sampling by semen collection (lines  304);  I conclusions, the author needs to reformulate (example 591-594).  Finally a suggest a deep  changes in conclusion.

My suggestion, due too many errors, contraditory results, and so other deficiencies, i suggest a major revision of this manuscript. If possible, i suggest the adaptation to a short communication after major revision

Author Response

Dear editor we corrected the manuscript addressing the reviewers' comments point-by-point and explaining the details of the revisions in our responses to their comments.

We hope that manuscript could be acceptable now for publication.

Below there are point-by-point responses to reviewer.

Reviewer # 2

This is a scientific domains were much work has been published. By example, see the references of this manuscript. Discussion is too big. It seems a bibliographic review. Many items were not treated in this manuscript. 4 pages is too much. This is not a review. Also conclusion needs to be shortened.

[Au]

We agree with this observation and accordingly, we considerably shortened the discussion and conclusion sections. We hope we successfully improved the soundness and readability of the manuscript.

This manuscript has many contradictory and unexpected results, namely of THI on progressive motility. Only in THI 5, there is a mild decrease.

[Au]

We agree with the observation, the reported results about spermatozoa motility are unexpected. Probably these results could be due to the fact that the considered THI classes were not too extreme. In fact, considering that the area in which the performance station is located is temperate, with only few extreme temperatures during the hottest months of the year, the number of bulls included in an extreme THI class would have been too low to perform a dedicated statistical analysis.

In this work, nothing is written concerning figures 2 and 3. if the several results are different (P<0.05.. I think that many values are not different (P<0.05) and the authors seems to indicate that they are different. I am somewhat confuse.

[Au]

We agree with the observation and try to describe those figures (updated as figure 3 and 4) more in detail. In particular we evaluate if, among all the single classes, statistically differences were revealed. Changes are visible from lines 352 to 360, form lines 365-377 and from lines 384-397.

Also some results are not published in tables (lines 249-251; 465-467).

[Au]

Due to the great number of tables and figures in the manuscript, we chose not to create a graph for each result. However, we inserted a new figure (Figure 2), which represents the trend of progressive motility through years. Numeration of the following figures was updated accordingly. Concerning the trend of sperm abnormalities, due to the non-significant effect of the year, we believe that a new figure is not necessary; thus, we specified that this information is simply described in the text by adding “data not shown” in line 262.

465-467:

[Au]

We agree with this observation and report the results of the statistical correlation among spermatozoa motility (total and progressive) and number of total abnormalities. Those results are visible in lines (269-271)

Line 506, Where are the results on the day of semen collection?

[Au]

According to the M&M, we did not analyse the results based on the THI on the day of semen collection: to better clarify, we modified the paper in lines 468-469.

Mass motility-How it was evaluated; for me mass motility is evaluated in a scale 0-5?

[Au]

According to this comment, we specified in lines 158-159 the method for the evaluation of total and progressive motility. Moreover, we replaced the term “mass” motility with the more appropriate “total” motility.

Lines 490-491- Any temperature was taken from the animals (see also lines, 490-491??)?; this is somewhat unexpected?

[Au]

The experimental design did not include body temperature measurement of bulls at the performance station, nor the measurement of specific parameter at the animal-level. Correlation between THI and alteration of clinical parameters in cattle is described in the review published by Hoffmann et al (https://doi.org/10.1016/j.biosystemseng.2019.10.017). Indeed, we indirectly used THI at time intervals before semen collection to estimate the risk of heat stress in the involved subjects.

Where are these results of correlation (line3 466-467)?

[Au]

See the upper response relative to lines 465-467.

Confuse lines 469-470?

[Au]:

We agree with the referee: the sentence was quite confusing. We re-phrased and briefly, we specified that Barth et al [38] found a positive correlation between normal and live sperm percentages. Changes are visible in lines [457-459]

Unexpected results (lines 546-548)?; this manuscript has many contradictory results.

[Au]

The reported increasing in spermatozoal motility during the evaluation of the effects of the increasing of THI classes at 30 and 10 days before semen collection, is an unexpected result because all the articles reported a decreasing pattern for motility after a thermal insult at any time during spermatogenesis.

The results obtained herein, were probably due to the paucity of bulls in extreme THI classes. It is likely that the animals were able to cope with the thermal insult.

In addition, even if the use of an animal-related indicator of heat stress could be used, in literature there are no objective thresholds for these parameters; differently thresholds for the different THI levels are wider described. We recognize that the result is contradictory to what can be found in the bibliography, but first of all is the only result contradictory, and correctly we think that we have to present it as detected and try to found possible explanation to it.

Also we see in discussion different methods, scrotal insulation (other manuscripts) vs present manuscript (see lines 548-549);

[Au]

In the discussion section we considered and compared our results with those from other authors which used the model of scrotal insulation. We agree with the Referee that methodology was different. However, the aim of the study was to evaluate the effect of the increasing temperature and humidity on semen traits, both from a short-term view (THI at -10 days to semen collection) and from a long-term perspective (-60 and -30 days intervals). This choice in order to examine the interactions between weather conditions and the entire spermatogenic cycle. Studies which adopted the scrotal insulation model used similar time intervals, thus being partly comparable.

On the other hand, as happen with many animal models, the scrotal insulation creates an artificial situation and exposes the animal to very high temperature and humidity, which reproduce a different situation in comparison to field condition; nevertheless, it is accepted as representative modification to that observed during environmental heat stress.

For this reason, we also included comparisons with other works, such as the report from Sabés-Alasina et al. and Llamas-Luceño et al. environmental heat stress.

Line 498, I don´t agree. So many publications in male; Many speculative, see lines 557-558 and 573;

[Au]

Accordingly, we deleted the sentence.

What means panting (line 494);

[Au]

We directly re-phrased all the paragraph: we hope we successfully improved the meaning of the sentence.

Some ortographic errors (lines 339, 537; 555; change sampling by semen collection (lines 304);

[Au]

We thank you for your suggestion: accordingly, we changed with “semen collection” through all the manuscript.

I conclusions, the author needs to reformulate (example 591-594). Finally suggest a deep changes in conclusion.

[Au]

We agree, and tried to reformulate and shorten the conclusion. We hope to have better summarized it focusing on the pivotal aims.

My suggestion, due too many errors, contradictory results, and soother deficiencies, i suggest a major revision of this manuscript. If possible, i suggest the adaptation to a short communication after major revision

[Au]

We respectfully point out to the reviewer, that the work object of this manuscript novel for the breed and for the geographical conditions considered; for this reason, it seems to us that the reduction of the manuscript to short communication is inadequate.

Round 2

Reviewer 2 Report

Some minor correction were made in this revised version. I will suggest the adaptation to a short communication after moderate corrections.

This manuscript has low scientific soundness because low number of  animals (see lines 484) were in the the higher TH4 and  THI 5, group, and it is not adequate to a solid statistical analysis. Results are totally unexpected and opposite to the scientific reports, namely progressive motility and THI (lines 478-480).

The effect of THI  on seminal parameters  should  have been held on the day of semen collection  to try to detect any influnce on THI on semen quality (See the bibliography). Line  495, 481 (Male insulation tecnhique) and its comparison with the technique used in this manuscript, were not the same, so discussion is poor;  Temperature evaluation were recommended on the animal, during this work. Unfortunately  it was not done.  Doing so, we may evaluation the THI on semen quality, and probably we may have surprises...; 

In this revises version i seem to see  higher number of ortographic errors (lines 355, 357, 386, 388 and 394) .

Due to my comments, this manuscript has low scientific soundness; so my suggestion is a new version adapted to short communication.

Author Response

Some minor correction were made in this revised version. I will suggest the adaptation to a short communication after moderate corrections.

This manuscript has low scientific soundness because low number of animals (see lines 484) were in the the higher TH4 and THI 5, group, and it is not adequate to a solid statistical analysis. Results are totally unexpected and opposite to the scientific reports, namely progressive motility and THI (lines 478-480).

[AU]

We would like to clarify that a total of 577 bulls were included in the present study. Even if the extreme THI included less numerous observations (see Table 3 in the manuscript, that is at least n = 118 for THI class 4 and from 64 to 102 bulls for THI class 5), interactions of THI classes on semen characteristics were still significant considering the 3 time intervals (see Table 4). In line 484 of discussion section, we affirmed that hypothesizing considering too extreme THI classes would have diminished the amount of bulls per each class. However, we did not proceed with this method, and we considered five THI classes, as described in the manuscript, with their statistically significant effect on semen characteristics. We agree that results are unexpected, but we are confident on their soundness, and that future investigations will focus on elucidating these discrepancies when comparing with literature.

The effect of THI on seminal parameters should have been held on the day of semen collection to try to detect any influence on THI on semen quality (See the bibliography). Line 495, 481 (Male insulation tecnhique) and its comparison with the technique used in this manuscript, were not the same, so discussion is poor; Temperature evaluation were recommended on the animal, during this work. Unfortunately it was not done. Doing so, we may evaluation the THI on semen quality, and probably we may have surprises...;

[Au]

The focus of the manuscript was to analyze the long-term effect of THI on seminal parameters, in order to evaluate the interaction of weather on the entire cycle of spermatogenesis. This because spermatogenesis in cattle takes approximately 60 days and because future sperm cells are susceptible to damage (oxidative stress, testis thermal insult…) during the process. Thus, the evaluation of THI only on the day of semen collection is scarcely informative; in fact, we cited works on scrotal insulation as method (model) to mimic the effect of environmental heat stress, which clearly demonstrated that the appearance of sperm defects in scrotal insulated bull start 10 to 15 days after thermal stress (see Vogler et al., 1993 and Rhaman et al., 2011). To further discuss difference between the two condition we added the following sentence: “Moreover, differently from scrotal insulation model, in our condition, thermal stress changes in the daytime with significant relief during the night”, which can also justify discrepancies reported with literature. We also evaluated separately the amount of primary and secondary defects (see Figure 4), as indicative of the phase of spermatogenesis involved.

Moreover, it should be considered that semen collection was performed under the direction of a veterinarian and that all bulls were submitted to clinical examination before semen collection, and symptoms of severe heat stress, which animals could not cope with (thus tachypnea, tachycardia, reduced feed intake, open mouth-breathing, etc.) were not observed.

In this revises version i seem to see higher number of ortographic errors (lines 355, 357, 386, 388 and 394).

[Au]

We thank you for your suggestion: accordingly, we corrected the errors at those lines. Moreover, we resubmit manuscript to language editing and further correction are reported in the text.

Due to my comments, this manuscript has low scientific soundness; so my suggestion is a new version adapted to short communication.

[Au]

We respect referee opinion, however we already justify our choice to not proceed in this sense in the previous round of responses.